# FoxO suppresses endoplasmic reticulum stress to inhibit growth of Tsc1-deficient tissues under nutrient restriction

Avantika Gupta, Hugo Stocker*

Institute of Molecular Systems Biology, ETH Zürich, Zürich, Switzerland

**Abstract** The transcription factor FoxO has been shown to block proliferation and progression in mTORC1-driven tumorigenesis but the picture of the relevant FoxO target genes remains incomplete. Here, we employed RNA-seq profiling on single clones isolated using laser capture microdissection from *Drosophila* larval eye imaginal discs to identify FoxO targets that restrict the proliferation of Tsc1-deficient cells under nutrient restriction (NR). Transcriptomics analysis revealed downregulation of endoplasmic reticulum-associated protein degradation pathway components upon *foxo* knockdown. Induction of ER stress pharmacologically or by suppression of other ER stress response pathway components led to an enhanced overgrowth of *Tsc1* knockdown tissue. Increase of ER stress in *Tsc1* loss-of-function cells upon *foxo* knockdown was also confirmed by elevated expression levels of known ER stress markers. These results highlight the role of FoxO in limiting ER stress to regulate *Tsc1* mutant overgrowth.

## Introduction

The phosphatidylinositol 3-kinase (PI3K)/Akt/mechanistic target of rapamycin complex 1 (mTORC1) network is the central regulator of cell growth, survival, and metabolism (*Manning and Toker, 2017*; *Saxton and Sabatini, 2017*). mTORC1 integrates growth factor signaling via Akt with the intracellular nutrient and energy status to promote anabolism and suppress catabolic processes (*Ben-Sahra and Manning, 2017*). The key regulatory function of the network is highlighted by its deregulation leading to human cancers (*Zhang et al., 2017*), and the evolutionary conservation across organisms including *Drosophila melanogaster* (*Engelman et al., 2006*; *Katewa and Kapahi, 2011*). In *Drosophila*, the major tumor suppressors of this network are phosphatase and tensin homolog (Pten) and the tuberous sclerosis complex (TSC) proteins Tsc1 and Tsc2. Pten counteracts the function of PI3K by dephosphorylating the second messenger phosphatidylinositol-3,4,5-triphosphate (PIP3) to phosphatidylinositol-4,5-biphosphate (PIP2) (*Klippel et al., 1997*), thereby inhibiting Akt activity (*Goberdhan et al., 1999*). The TSC complex inactivates mTORC1 downstream of Akt (*Potter et al., 2002*) via its GTPase activating protein (GAP) activity towards Rheb (*Saucedo et al., 2003*; *Stocker et al., 2003*; *Zhang et al., 2003*). Another major downstream target of Akt is the transcription factor Forkhead box O (FoxO) (*Jünger et al., 2003*). However, the role of FoxO as a bona fide tumor suppressor is not established since it can also function as tumor-promoting depending on the biological context (*van Doeselaar and Burgering, 2018*). This underscores the importance of understanding the regulation of FoxO and the downstream signaling in a particular context of interest (*Brown and Webb, 2018*).

Using an early tumorigenesis model in *Drosophila*, we have shown that the loss of Tsc1 leads to an enhanced overgrowth of eye imaginal discs in larvae raised under nutrient restriction (NR) (*Nowak et al., 2018*). This overgrowth is primarily hypertrophic in nature. Loss of Tsc1 induces hyperactivation of mTORC1 and an S6K-dependent negative feedback to insulin signaling. This causes downregulation of Akt and an increased nuclear localization of FoxO, resulting in growth

*For correspondence: stocker@imsb.biol.ethz.ch

Competing interests: The authors declare that no competing interests exist.

inhibition. Suppression of FoxO function in *Tsc1* mutant cells results in massively overgrown tissues that show ectopic differentiation, demonstrating a role of FoxO in regulating proliferation and differentiation of these hyperactive mTORC1 cells. However, the specific FoxO targets crucial for growth regulation remained elusive.

The *Drosophila* eye imaginal disc is composed of different populations of mitotically active and differentiating cells. Given the context and cell-type specificity of FoxO (*Webb et al., 2016*), investigation for its targets is required to be performed in specific cells under conditions being studied. The laser capture microdissection (LCM) technique has emerged as a useful tool for isolation of distinct cells of interest (*Iyer and Cox, 2010*; *Vicidomini et al., 2010*) but a comprehensive analytical study has not been described previously in *Drosophila*. Here, we use transcriptome profiling of single mutant clones isolated using LCM from larval eye imaginal discs. Our results identify downregulation of ER stress as a key mechanism downstream of FoxO to inhibit *Tsc1* mutant overgrowth under NR.

## Results

### Transcriptomics analysis to identify FoxO targets in *Tsc1* mutant cells under NR

FoxO restricts the growth of tissues with high mTORC1 activity under NR (*Nowak et al., 2018*). To identify FoxO targets essential for this growth regulation, we performed RNA-seq transcriptome profiling of *Tsc1* mutant cells, with or without *foxo* knockdown, using our previously described model of early tumorigenesis in *Drosophila* larval eye imaginal discs. The mutant cells were isolated in a spatially and temporally controlled manner to address the context specificity and high number of FoxO targets (*Webb et al., 2016*). To profile strictly the mitotically active cells and to avoid false-positive FoxO targets from different cell types, the LCM technique was used to isolate single clones from the mitotically active part anterior to the morphogenetic furrow in the eye disc (spatial control) of wandering third instar larvae (temporal control). The MARCM (*Lee and Luo, 2001*) and Gal80^ts systems were combined to gain a temporal control on the expression of *foxo-RNAi* specifically in mutant clones (*Figure 1A*). Evaluating the nuclear FoxO antibody staining in *Tsc1* mutant cells (*Manning et al., 2005*) in time-course experiments revealed that a 12 hr shift was sufficient to achieve an adequate *foxo* knockdown (*Figure 1B and B'*). The knockdown of *foxo* for a short duration limited the detection of secondary transcriptional targets, as illustrated earlier (*Gan et al., 2010*).

Using the above-mentioned techniques, four single clones each were captured for isogenic control and *Tsc1* mutant cells, with or without *foxo* knockdown, from larvae raised on normal food and NR. *foxo* knockdown was achieved by shifting larvae to the restrictive temperature (30℃) 12 hr before dissection, and complementary control clones were isolated from siblings raised at 25℃. Analysis was done on 29 samples that passed quality control. Clustering on normalized gene counts showed high variability among the biological replicates (*Figure 1—figure supplement 1*). This can be explained by the small starting material for library preparation (*Bhargava et al., 2015*) and an absence of pooling of multiple samples to average the individual organismal differences. Interestingly, a reduction in *foxo* transcripts was not observed in the knockdown samples (*Figure 1—figure supplement 1*). A similar partial loss of transcripts accompanied by a significant loss of protein (*Figure 1—figure supplement 1*) was observed in Kc167 cells, suggesting the presence of a transcriptionally-activating feedback upon *foxo* knockdown or a direct effect on the translational capacity (*Doench et al., 2003*). Representative images of isolated *Tsc1* mutant clones with and without *foxo* knockdown are presented to support that the processed cell populations were not contaminated with GFP-positive non-clonal tissue (*Figure 1—figure supplement 2*). This is further substantiated by the presence of 100% mutant transcripts (*Tsc1^Q87X* where codon CAG at position 3R:24,130,984 is substituted by TAG) in the *Tsc1* mutant clones (example in *Figure 1—figure supplement 2*). Finally, gene set enrichment analysis (GSEA) (*Subramanian et al., 2005*) showed enrichment of genes upregulated in *Tsc1* mutant discs (*Harvey et al., 2008*) as well as FoxO targets from adipose tissue and muscle (*Teleman et al., 2008*) in corresponding lists of differentially expressed genes (*Figure 1—figure supplement 2*).

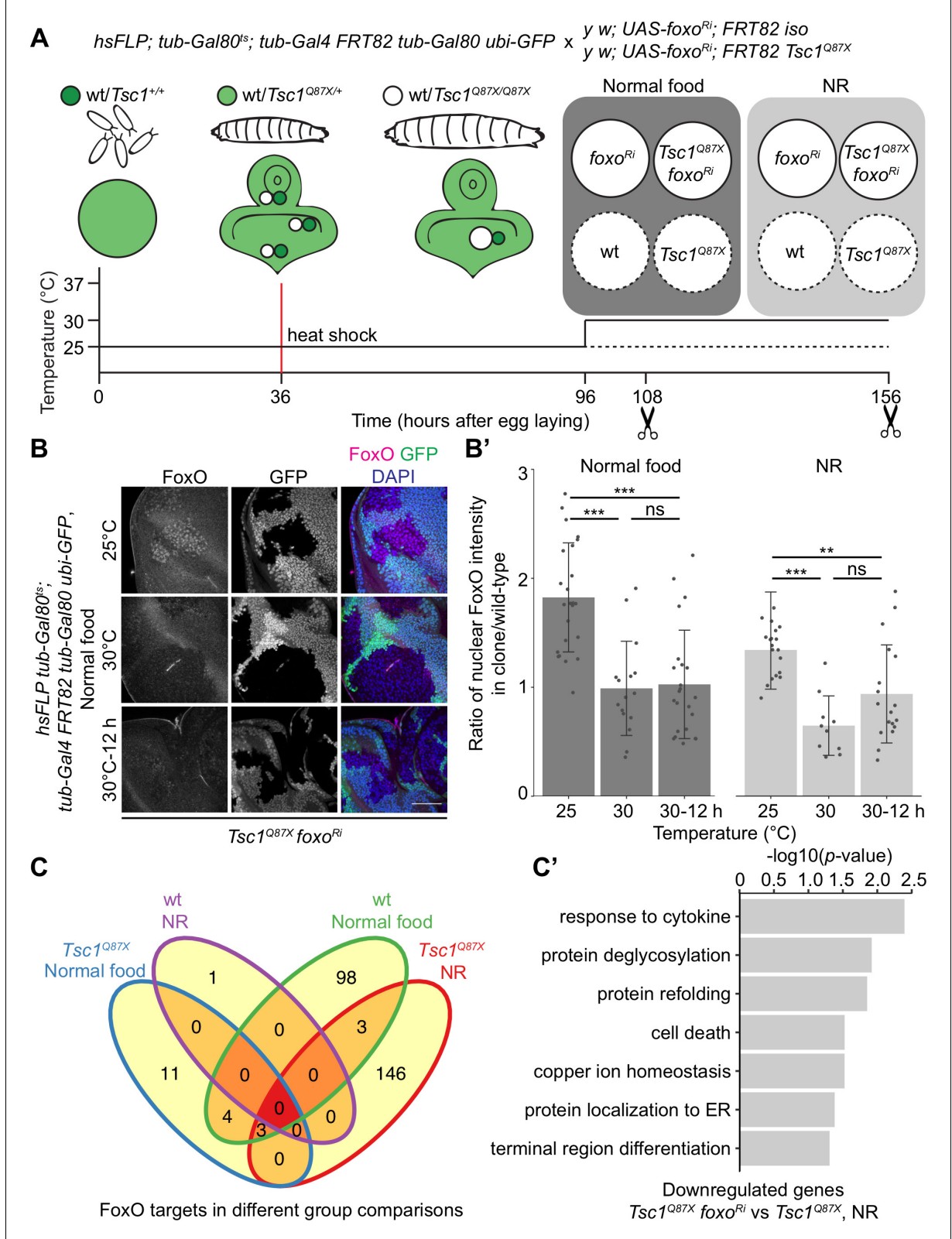

**Figure 1.** Transcriptomics analysis to identify FoxO targets in *Tsc1* mutant cells. (**A**) Schematic of the experimental setup to generate temporal knockdown of *foxo* in wild-type or *Tsc1* mutant clones, and isolation of single clones using LCM at 108 hr after egg laying (AEL) from normal food and 156 hr AEL from NR. Solid or dashed lines represent clones isolated from larvae shifted to 30°C for 12 hr or maintained at 25°C, respectively. (**B**) FoxO staining of eye imaginal discs with *Tsc1* mutant clones dissected from larvae raised on normal food at 25°C, 30°C or shifted from 25°C to 30°C for 12 hr.

*Figure 1 continued on next page*

*Figure 1 continued*

Clones are negatively marked by GFP, and DAPI stains nuclei. Scale bar = 50 µm. (B') Quantification of ratio of nuclear FoxO intensity in *Tsc1* mutant clone over wild-type from larvae raised on normal food or NR at temperatures described in B. n > 9. Data are represented as mean ± SD. **p<0.01, ***p<0.001 and ns = not significant. (C) Venn diagram depicting number of genes, upregulated and downregulated, between all conditions tested. p<0.0025 and FDR < 0.2. (C') Gene ontology analysis of the downregulated genes in *Tsc1* mutant cells upon *foxo* knockdown as compared to *Tsc1* mutant cells under NR.

The online version of this article includes the following figure supplement(s) for figure 1:

**Figure supplement 1.** Validation of RNA-seq data.
**Figure supplement 2.** Contamination status of isolated clones.

Differential gene expression analysis of *Tsc1* mutant cells, with or without *foxo* knockdown, under NR revealed 24 downregulated and 128 upregulated genes (ANOVA, logFC > 0.8 and FDR < 0.2, *Figure 1C*) in the absence of *foxo*. Gene Ontology analysis of the downregulated genes showed enrichment in various biological processes such as protein deglycosylation, protein refolding, and protein localization to ER (*Figure 1C'*), and these genes have been further validated (see below). From the 128 upregulated genes, more than half code for small nucleolar RNAs (snoRNAs), and associated small nuclear RNAs (snRNAs).

## In vivo validation of downregulated genes in *Tsc1* mutant cells upon *foxo* knockdown identifies endoplasmic reticulum-associated protein degradation (ERAD) pathway components as top hits

The downregulated genes in *Tsc1* mutant cells upon *foxo* knockdown were validated functionally for their regulation of *Tsc1* mutant overgrowth, similar to FoxO, in an in vivo RNAi assay. Different RNAi lines for 24 candidates (*Figure 2A*) were crossed to a tester line utilizing the eye-specific FLP-out

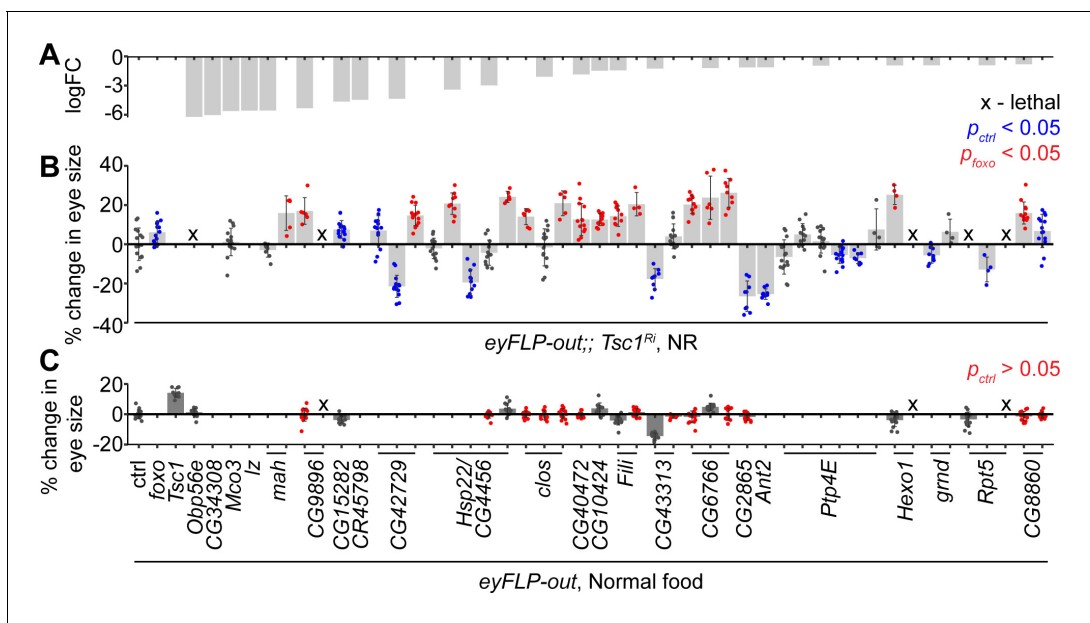

**Figure 2.** In vivo RNAi assay to validate downregulated genes in *Tsc1* mutant cells upon *foxo* knockdown. (A) log fold change (FC) of downregulated genes in *Tsc1* mutant cells with *foxo* knockdown versus *Tsc1* mutant cells under NR. (B) Percentage change in adult eye size upon knockdown of candidate genes in *Tsc1* knockdown background under NR. n > 3. Blue or red points represent a significant (p<0.05) change as compared to ctrl or *foxo* knockdown, respectively. The RNAi lines are ordered as in the Materials and methods. (C) Percentage change in adult eye size upon knockdown of candidate genes in wild-type background on normal food. n > 9. Red points represent a non-significant (p>0.05) change as compared to ctrl. Data are represented as mean ± SD.

The online version of this article includes the following figure supplement(s) for figure 2:

**Figure supplement 1.** In vivo RNAi assay of downregulated candidates with strong *Tsc1* knockdown and representative phenotypes of top hits.
**Figure supplement 2.** Loss of CG6766 enhances ER stress in *Tsc1* mutant cells.

system (*Struhl and Basler, 1993*) with *Tsc1-RNAi. foxo* knockdown led to a significant size increase of *Tsc1* knockdown eyes as compared to the control (*Figure 2B*). Co-knockdown of eight candidate genes using different RNAi lines consistently enhanced the overgrowth of *Tsc1* knockdown eyes. Candidates that scored positive were also assessed in a wild-type background to eliminate additive growth effects with loss of *Tsc1* (*Figure 2C*). Knockdown of four candidate genes did not affect eye size (statistically not significant data highlighted in red) in the wild-type background, indicating that the growth effect was specific to the loss of *Tsc1*. A weak knockdown transgene of *Tsc1* was used in the above assays because a combined knockdown of *Tsc1* and *foxo* is larval lethal (*Nowak et al., 2018*). The effect on eclosion upon knockdown of the candidates with *Tsc1* using the strong knockdown allele was also quantified (*Figure 2—figure supplement 1*).

The top candidate from all genetic tests was the uncharacterized gene CG6766, encoding an ortholog of the mammalian endoplasmic reticulum lectin 1 (ERLEC1). It is predicted to localize to ER/Golgi and is considered to be a component of the ubiquitin-dependent ERAD pathway (*Huang et al., 2018*). Another candidate that is involved in protein processing in the ER, CG8860, also scored positively in the in vivo assays, suggesting a role for ER homeostasis in determining *Tsc1* mutant overgrowth. CG8860 is orthologous to the mammalian SEC61 translocon gamma subunit (SEC61G). The knockdown phenotypes of the candidates from the in vivo assays are shown in *Figure 2—figure supplement 1*.

Since the function of the top candidate in ER homeostasis is not characterized in *Drosophila*, the induction of ER stress response in cells mutant for *CG6766* was investigated. A mutant allele carrying a piggyBac insertion with a splice-acceptor site and stop codons in all frames in the first intron of the gene (*Schuldiner et al., 2008*) was used. Calnexin is an ER molecular chaperone and a $Ca^{2+}$ homeostasis regulator (*Arruda and Hotamisligil, 2015*) that is localized to the ER membrane (*Xiao et al., 2017*). Eye imaginal discs with control or mutant clones dissected from larvae raised under normal feeding conditions were immunostained using an antibody against the *Drosophila* calnexin homolog, Cnx99A. Homozygous mutant clones showed similar staining patterns compared to the surrounding wild-type tissue and the isogenic control clones (*Figure 2—figure supplement 2*). To examine the effect on ER stress response in Tsc1-deficient cells, the transcript levels of known ER stress reporters, spliced-*Xbp1* and *Hsc-70–3* (see below), were examined. Control or *CG6766* knockdowns were generated in wild-type, weak *Tsc1* knockdown, and strong *Tsc1* knockdown backgrounds. Quantitative reverse-transcription PCR (RT-qPCR) analysis on eye imaginal discs revealed no change in the expression levels of spliced-*Xbp1* or *Hsc-70–3* in the wild-type background upon knockdown of *CG6766*. By contrast, the expression levels of both ER stress reporters were elevated in *Tsc1* knockdown backgrounds (*Figure 2—figure supplement 2*), indicating that CG6766 is limiting to maintain ER homeostasis in *Tsc1* mutant cells. The lack of ER stress response in cells mutant for CG6766 suggests a (partially) redundant function, and may also explain why CG6766 (as well as CG8860) has not been characterized so far.

## Induction of ER stress enhances the growth of *Tsc1* mutant eyes

To confirm the role of ERAD in the regulation of growth of *Tsc1* mutant tissues, ERAD components other than the genes identified by the transcriptomics analysis were knocked down using the same genetic assay (*Figure 2B*). The genes were functionally classified according to the KEGG database (dme04141, *Figure 3A*). Knockdown of several ERAD pathway components led to an increase in *Tsc1* knockdown eye size (*Figure 3B*). Particularly, inhibition of the core components of the ubiquitin ligase complex and those involved in the retrotranslocation of terminally misfolded proteins from ER to cytosol resulted in an enlargement of eye size. Of note, CG13426, which encodes an ortholog of SEC61G similar to the candidate from the transcriptomics analysis, CG8860, also led to an enhancement of the *Tsc1* knockdown overgrowth.

A reduced efficiency of ERAD lowers the capacity of the ER to remove misfolded proteins. In mammals, accumulation of misfolded proteins in the ER causes activation of the three signal transduction pathways: PERK, IRE1$\alpha$ and ATF6, together referred to as the unfolded protein response (UPR) (*Wang and Kaufman, 2014*). These signal to induce apoptosis in case of chronic ER stress, or maintain ER homeostasis by upregulating expression of several ERAD pathway genes. The ERAD pathway also links the ubiquitin-proteasome system to ER protein quality control as its substrates are transported into the cytosol for degradation by the proteasome (*Ruggiano et al., 2014*). To analyze if downregulation of UPR or proteasomal degradation pathway components would affect *Tsc1*

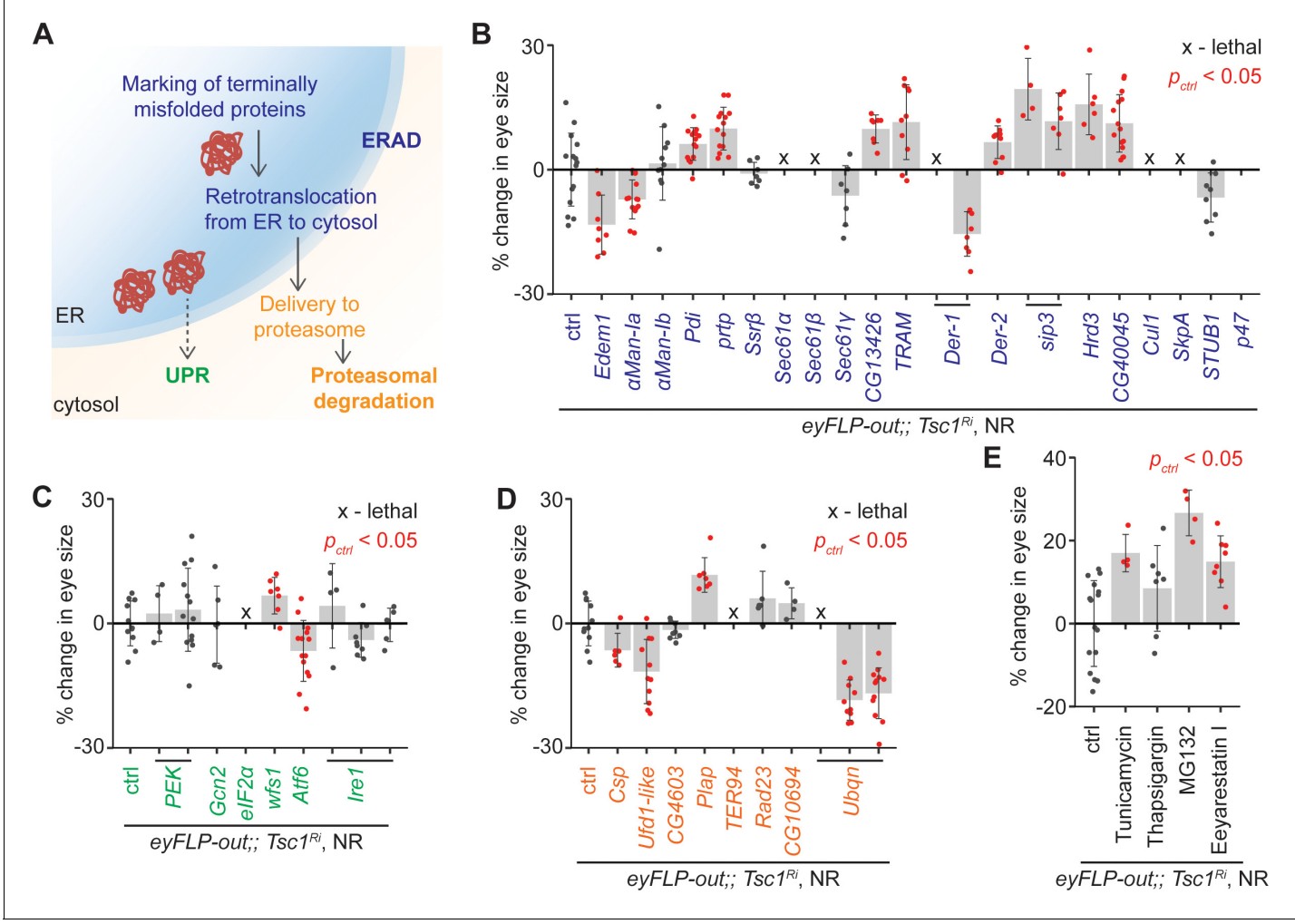

**Figure 3.** Upregulation of ER stress increases *Tsc1* knockdown overgrowth. (**A**) Schematic depicting involvement of genes tested in (**B**), (**C**) and (**D**) in different ER stress regulation mechanisms according to the KEGG database. Blue, green and orange denote components of ERAD, UPR and proteasomal degradation pathways, respectively. Percentage change in adult eye size upon knockdown of (**B**) ERAD, (**C**) UPR and (**D**) proteasomal degradation pathway components in *Tsc1* knockdown background under NR. Note that experiments in (**C**) and (**D**) were performed together and therefore have the same control. (**E**) Percentage change in *Tsc1* knockdown adult eye size upon addition of the indicated small molecule inhibitors to the fly food under NR. n > 3. Data are represented as mean ± SD. Red points represent a significant (p<0.05) change as compared to ctrl.

knockdown overgrowth similar to the suppression of ERAD pathway genes, the major components of these pathways were also assessed. Knockdown of UPR or proteasomal degradation pathway genes did not display a consistent effect on *Tsc1* knockdown overgrowth (*Figure 3C and D*), indicating that the effects might be specific to ERAD or that the manipulations are detrimental (proteasomal degradation pathway components).

Furthermore, the consequence of the induction of ER stress on *Tsc1* knockdown overgrowth by using non-genetic means was tested. The inhibition of N-linked glycosylation by Tunicamycin (*Kuo and Lampen, 1974*), $Ca^{2+}$ ATPase by Thapsigargin (*Lytton et al., 1991*), proteasome by MG132 (*Lee and Goldberg, 1998*), and ERAD by Eeyarestain I (*Fiebiger et al., 2004*) cause ER stress. Addition of these small molecule inhibitors to the fly food, except Thapsigargin, resulted in an enhancement of *Tsc1* knockdown eye size (*Figure 3E*), consistent with a regulation of Tsc1-deficient growth by the ER stress response status.

## Loss of FoxO function upregulates ER stress in *Tsc1* mutant cells under NR

Since both FoxO suppression and ER stress induction caused an escalation in *Tsc1* mutant overgrowth, the ER stress status upon *foxo* loss in *Tsc1* mutant cells was examined. Eye imaginal discs with ctrl, *foxo*, *Tsc1* or *Tsc1 foxo* mutant clones dissected from larvae raised under NR were immunostained with calnexin (*Figure 4A*). The staining was stronger in *Tsc1* and *Tsc1 foxo* mutant clones as compared to the surrounding wild-type tissue, suggesting an upregulation of Cnx99A levels or a change in ER morphology in these cells.

Increased ER stress in *Tsc1* loss-of-function cells has been established (*Ozcan et al., 2008*). Therefore, to compare the status upon loss of *foxo*, the cells were probed for transcript levels of a known ER stress reporter, the resident ER chaperone *Hsc-70–3* or *Grp78* (*Elefant and Palter, 1999*). RT-qPCR analysis of *Tsc1* or *Tsc1 foxo* mutant eye imaginal discs dissected from larvae raised under

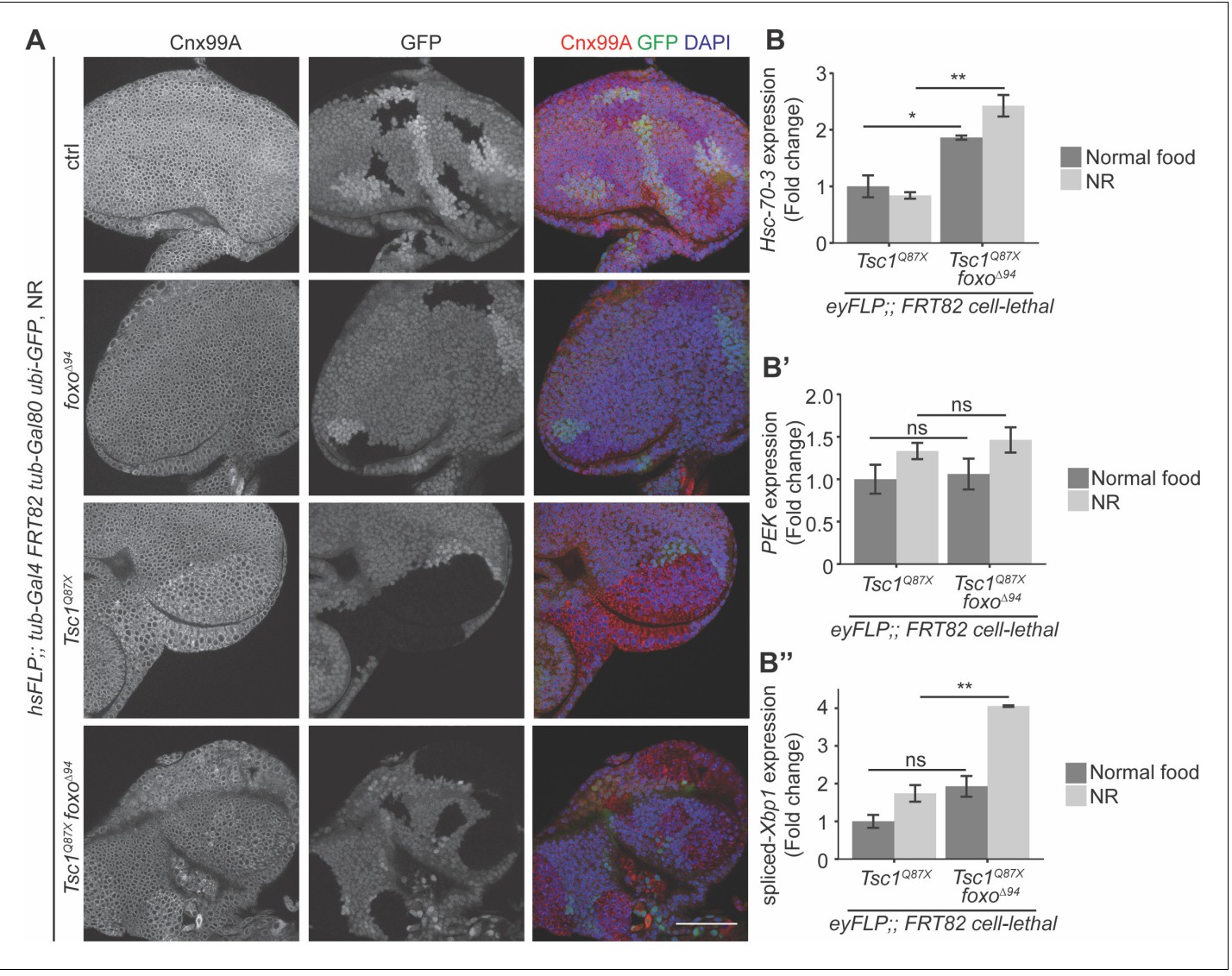

**Figure 4.** Loss of *foxo* enhances ER stress in *Tsc1* mutant cells. (A) Calnexin 99A staining of eye imaginal discs with ctrl, *foxo*, *Tsc1* or *Tsc1 foxo* mutant clones dissected from larvae raised under NR. Clones are marked by the absence of GFP; DAPI stains nuclei. Scale bar = 50 μm. (B–B'') RT-qPCR analysis of (B) *Hsc-70–3*, (B') *PEK* and (B'') spliced-*Xbp1* on *Tsc1* or *Tsc1 foxo* mutant eye imaginal discs dissected from larvae raised under normal food or NR. n = 2 with triplicates. Data are represented as mean ± SD. *p<0.05, **p<0.01 and ns = not significant.

The online version of this article includes the following figure supplement(s) for figure 4:

**Figure supplement 1.** FoxO suppresses Ire1 pathway activation in *Tsc1* mutant cells under NR.

normal food or NR showed elevated *Hsc-70–3* transcript levels upon *foxo* loss (*Figure 4B*), indicating an enhancement of ER stress. Induction of UPR was also checked by analyzing the two conserved branches in *Drosophila*: PEK (PERK) and Ire1 (*Mori, 2009*). *PEK* transcript levels were not changed upon *foxo* loss in *Tsc1* mutant eye discs (*Figure 4B'*). Ire1 signals via splicing of the transcription factor X-box binding protein 1 (Xbp1) (*Yoshida et al., 2001*). The expression levels of spliced-*Xbp1* were significantly increased in *Tsc1 foxo* mutant eye discs, specifically under conditions of NR (*Figure 4B''*). The enrichment of spliced-*Xbp1* was further confirmed in *Tsc1 foxo* knockdown clones under NR using an in vivo reporter (*Sone et al., 2013*; *Figure 4—figure supplement 1*). This illustrates the capacity of FoxO to regulate the ER stress status in *Tsc1* mutant cells.

## Discussion

The transcription factor FoxO has a distinct activity in different biological contexts. In cells with loss of *Tsc1* function, it suppresses cellular growth and proliferation. In this study, we performed RNA-seq transcriptomics analysis on single mutant clones isolated by LCM to identify critical regulators of *Tsc1* mutant tissue growth upon loss of FoxO. Earlier studies (*Mauthner et al., 2014*; *Spletter et al., 2007*) have described the use of LCM for RNA isolation from optimal cutting temperature compound-embedded tissues, but this is the first report, to our knowledge, illustrating the use of this technique with living imaginal disc cells in *Drosophila* that was not based on pooling of multiple biological samples.

The majority of the genes differentially upregulated in *Tsc1* mutant cells upon *foxo* knockdown consisted of snoRNAs and snRNAs. snoRNAs are single-stranded non-coding RNAs that function primarily in the posttranscriptional modification of ribosomal RNAs and snRNAs to regulate ribosome biogenesis and function, RNA editing, gene silencing, and alternative splicing (*Bratkovič and Rogelj, 2011*). snoRNA genes were shown to be directly bound and activated by Myc in *Drosophila* S2 cells (*Herter et al., 2015*). On the other hand, the expression of *myc* is suppressed by FoxO, at least in *Drosophila* muscles (*Teleman et al., 2008*). The same study reported that mTORC1 regulates Myc activity by controlling the amount of protein bound to promoters of target genes. A similar mechanism might be at play in the *Tsc1* mutant cells of the eye imaginal discs. Upon knockdown of *foxo*, the inhibition of *myc* expression by FoxO is alleviated under NR, and Myc gets activated downstream of hyperactive mTORC1. This can lead to a subsequent upregulation of snoRNAs, which helps in biomass production in these rapidly growing cells. Indeed, emerging evidence suggests a correlation between snoRNAs and sustained proliferation, invasion, metastasis, and angiogenesis in different cancers (*Liang et al., 2019*).

From among the downregulated genes, ER stress response pathway components were validated as essential regulators of *Tsc1* mutant overgrowth. There is a significant interplay between the PI3K/Akt/mTORC1 signaling network and ER stress pathways (*Appenzeller-Herzog and Hall, 2012*). Induction of ER stress results in the suppression of insulin receptor signaling linking ER stress to insulin resistance (*Ozcan et al., 2004*). Similarly, PEK-driven regulation of FoxO contributes to this resistance in *Drosophila* (*Zhang et al., 2013*). In a study on cardiac dysfunction, uncontrolled activation of Akt repressed ER stress (*Dong et al., 2013*). Hyperactivation of mTORC1 activity, particularly via loss of the TSC complex function, leads to increased ER stress and UPR pathway upregulation (*Ozcan et al., 2008*). In turn, ER stress inhibits Akt and mTORC1 activity to induce autophagy (*Qin et al., 2010*). Our identification of putative ERAD components governing *Tsc1* mutant tissue growth adds another interaction node to this pathway crosstalk. The absence of canonical ERAD components in the transcriptomics data prompted us to consider the possibility of an independent mechanism for the degradation of misfolded proteins, as shown previously in *C. elegans* (*Safra et al., 2014*). However, enhancement of the *Tsc1* loss-of-function overgrowth upon downregulation of the core ERAD components implies an important role of the Hrd3-E3 ubiquitin ligase complex in this growth process.

The results presented here demonstrate a critical function of FoxO in limiting ER stress to restrain *Tsc1* mutant overgrowth under NR. Given that FoxO restricts *Tsc1* mutant tissue growth at the level of proliferation, a lack of known proliferation-affecting genes among the differentially regulated genes is intriguing. This suggests that the inhibitory effect on proliferation by FoxO might be indirect, possibly via ER stress response pathways (*Vandewynckel et al., 2013*). PERK reduces oxidative DNA damage and checkpoint activation to increase proliferation of cancer cells (*Bobrovnikova-*

*Marjon et al., 2010*). A role for the ER chaperone GRP78 in regulating tumor cell proliferation by various means has also been described (*Lee, 2014*). IRE1α has been shown to promote cell proliferation by regulating expression of cyclin A1 through XBP-1 splicing (*Thorpe and Schwarze, 2010*). Our results indeed demonstrate a heightened expression of *Hsc-70–3* and spliced-*Xbp1* in *Tsc1* mutant cells upon *foxo* loss under NR, hinting at a possible approach to reinforce proliferation in these cells. PEK might also be regulated at a level other than transcription. The exact molecular mechanisms of this regulation remain to be elucidated.

Aberrant ER stress activation has also been recorded during various aspects of cancer development (*Avril et al., 2017*; *Wang and Kaufman, 2014*). Considerable efforts have been made to develop therapeutic strategies to target the dependency of tumor cells on ER stress response pathways, or to substantially upregulate stress to cause cell death (*Hetz et al., 2013*). Specifically, in the case of loss of *Tsc1/2* function, use of ER stress-inducing agents has displayed sensitization of mutant cells to individual treatments in vitro (*Kang et al., 2011*), or combination treatments in a mouse xenograft tumor model (*Li et al., 2015*). Since the ER stress response can have both pro-survival and pro-apoptotic effects (*Ma and Hendershot, 2004*), it is crucial to characterize the signaling landscape of individual tumors before therapeutic intervention. For instance, our results demonstrate that increasing ER stress led to an enhancement of *Tsc1* mutant overgrowth. A similar overgrowth mechanism can be proposed for *Tsc1 foxo* mutant cells. Suppression of *foxo* represses the ERAD pathway activation, which combined with the augmented protein synthesis in *Tsc1* mutant cells results in a state of high ER stress. This new adaptive state is beneficial for proliferation and confers a growth advantage to *Tsc1 foxo* mutant cells. The exploitation of this process in tumors with escalated mTORC1 activity that exhibit loss of FoxOs can extend the repertoire of available treatment options and needs further exploration.

# Materials and methods

## Fly food and fly lines

All crosses were maintained at 25°C on normal fly food unless otherwise stated. Normal fly food is composed of 100 g fresh yeast, 55 g cornmeal, 10 g wheat flour, 75 g sugar, 8 g bacto agar, and 1.5% antimicrobial agents (33 g/L nipagin and 66 g/L nipasol in ethanol) in 1 L water. NR food was prepared by reducing the amount of yeast to 10% with 1.5% nipagin (100 g/L in ethanol). The following small molecule inhibitors were added to the NR food before pouring into vials for the drug assay: DMSO (control, 0.1%, Carl Roth Cat#A994.1), Tunicamycin (12 μM, Sigma-Aldrich Cat#T7765), Thapsigargin (2 μM, Sigma-Aldrich Cat#T9033), MG132 (50 μM, Sigma-Aldrich Cat#C2211) and Eeyarestatin I (1 mM, Sigma-Aldrich Cat#E1286).

Analysis on adult flies was done on females in all experiments. Larvae were not selected for sex. The age of adults and larvae are mentioned in the corresponding Figure legends and Materials and methods.

The following fly lines were used: *hsFLP* (FBtp0001101 *Struhl and Basler, 1993*), *eyFLP Act>CD2>Gal4* (*Baumgartner et al., 2013*), *eyFlp gl-LacZ* (*Newsome et al., 2000*), *Act>CD2>Gal4 UAS-RFP* (BDSC Cat# 30558), *FRT82 iso* (*Xu and Rubin, 1993*), *FRT82 foxo$^{Δ94}$* (FBal0269838 *Slack et al., 2011*), *FRT82 Tsc1$^{Q87X}$* (FBal0123965 *Tapon et al., 2001*), *FRT82 Tsc1$^{Q87X}$ foxo$^{Δ94}$* (*Nowak et al., 2018*), *FRT82 cl w$^+$*(*Newsome et al., 2000*), *tub-Gal4 FRT82 tub-Gal80 ubi-GFP* (this paper), *tub-Gal80$^{ts}$* (BDCS Cat #7108), RNAi of CG1315 (VDRC ID# 47097), RNAi of *foxo* (VDRC ID# 1077896), RNAi of *Tsc1* (BDSC Cat# 35144), RNAi of *Tsc1* (strong, BDSC Cat# 31039), RNAi of *Obp56e* (VDRC ID# 3355), RNAi of *Mco3* (VDRC ID# 43288), RNAi of *mah* (VDRC ID# 1571, BDSC Cat# 53701), RNAi of CG9896 (VDRC ID# 107926, BDSC Cat# 42587), RNAi of CG15282 (VDRC ID# 7712), RNAi of CG42749 (VDRC ID# 28574, VDRC ID# 51132, VDRC ID# 51133), RNAi of *Hsp22/ CG4456* (VDRC ID# 43632, BDSC Cat# 41709, BDSC Cat# 51397, VDRC ID# 49795, VDRC ID# 49796), RNAi of *clos* (VDRC ID# 104142, VDRC ID# 108711, BDSC Cat# 52966), RNAi of CG40472 (VDRC ID# 109239), RNAi of CG10424 (VDRC ID# 39667), RNAi of *Fili* (VDRC ID# 106055, BDSC Cat# 28568), RNAi of CG43313 (VDRC ID# 106610, BDSC Cat# 53990), RNAi of CG6766 (VDRC ID# 38035, BDSC Cat# 42924, BDSC Cat# 55745), RNAi of CG2865 (BDSC Cat# 43165), RNAi of *Ant2* (VDRC ID# 102533), RNAi of *Ptp4E* (VDRC ID# 1012, VDRC ID# 1013, VDRC ID# 4297, VDRC ID# 27232, BDSC Cat# 38369, BDSC Cat# 60008), RNAi of *Hexo1* (VDRC ID# 102247, BDSC Cat#

67312), RNAi of *grnd* (VDRC ID# 43454, VDRC ID# 104538), RNAi of *Rpt5* (VDRC ID# 105133, BDSC Cat# 32422, BDSC Cat# 53886), RNAi of *CG8860* (VDRC ID# 102745, BDSC Cat# 60127), RNAi of *Edem1* (VDRC ID# 6923), RNAi of *α-Man-Ia* (VDRC ID# 100654), RNAi of *α-Man-Ib* (VDRC ID# 4419), RNAi of *Pdi* (VDRC ID# 23359), RNAi of *prtp* (VDRC ID# 106924), RNAi of *Ssrβ* (VDRC ID# 12101), RNAi of *Sec61α* (VDRC ID# 109660), RNAi of *Sec61β* (VDRC ID# 107784), RNAi of *Sec61γ* (VDRC ID# 100603), RNAi of *CG13426* (VDRC ID# 107528), RNAi of *TRAM* (VDRC ID# 39187), RNAi of *Der-1* (VDRC ID# 44210, VDRC ID# 44211), RNAi of *Der-2* (VDRC ID# 108440), RNAi of *sip3* (BDSC Cat# 50609, BDSC Cat# 61344), RNAi of *Hrd3* (BDSC Cat# 62368), RNAi of *CG40045* (VDRC ID# 109167), RNAi of *PEK* (BDSC Cat# 35162, BDSC Cat# 42499), RNAi of *Gcn2* (VDRC ID# 32664), RNAi of *eIF2α* (VDRC ID# 104562), RNAi of *Atf6* (VDRC ID# 36504), RNAi of *wfs1* (VDRC ID# 108932), RNAi of *Ire1* (BDSC Cat# 36743, BDSC Cat# 62156, BDSC Cat# 35253), RNAi of *Cul1* (VDRC ID# 108558), RNAi of *SkpA* (VDRC ID# 107815), RNAi of *STUB1* (VDRC ID# 107447), RNAi of *p47* (VDRC ID# 107148), RNAi of *Csp* (VDRC ID# 34168), RNAi of *TER94* (VDRC ID# 24354), RNAi of *Ufd1-like* (VDRC ID# 24700), RNAi of *CG4603* (VDRC ID# 21894), RNAi of *Plap* (VDRC ID# 110732), RNAi of *Ubqn* (VDRC ID# 106050, VDRC ID# 47447, VDRC ID# 47448), RNAi of *CG10694* (VDRC ID# 100212), RNAi of *Rad23* (VDRC ID# 104354), *CG6766*[140290] (Kyoto *Drosophila*Genetic Resource Center ID# 140290), *UAS-Xbp1-EGFP* (BDSC Cat# 60730).

## Cell lines
Kc167 (female) cells were obtained from *Drosophila* Genomics Resource Center (Cat#1). Cells were cultured in Schneider's medium (Gibco) with 10% Fetal Bovine Serum and 500 U/mL Penicillin-Streptomycin at 24°C.

## Laser capture microdissection
Eye imaginal discs were dissected in HBSS and mounted in a 0.5 µL drop of HBSS on FrameSlide PET (Carl Zeiss AS). Single clones of approximately 50 cells were microdissected using a Zeiss PALM-Microdissection system and captured into the caps of AdhesiveCap (Carl Zeiss AS) microcentrifuge tubes. The samples were frozen in liquid nitrogen and stored at −80°C until RNA isolation.

## RNA-seq: library preparation and analysis
RNA isolation and cDNA library preparation from the LCM samples were done according to the Ovation SoLo RNA-seq Systems kit (NuGEN Cat#0502–32). The sequencing was performed at the Functional Genomics Center Zurich on a HiSeq 2500 System (Illumina). Reads were mapped to the *Drosophila* genome (BDGP6) using STAR (*Dobin et al., 2013*), counted with featureCounts (*Liao et al., 2014*), and differentially expressed genes were analyzed using edgeR (*Robinson et al., 2010*). Gene Ontology analysis was performed using DAVID Bioinformatics Resources website (*Huang et al., 2009*), and GSEA (*Subramanian et al., 2005*) was run with default parameters to calculate enrichment with genes upregulated in *Tsc1* mutant discs (*Harvey et al., 2008*) as well as FoxO targets from adipose tissue and muscle (*Teleman et al., 2008*) in corresponding lists of differentially expressed genes. The RNA-seq transcriptomics data have been deposited to NCBI's Gene Expression Omnibus.

## Cross setup and clone induction
Flies were crossed for two days on normal food. For assays using imaginal discs, the flies were transferred to egg laying chambers and allowed to lay eggs on agar plates containing yeast overnight. For clone induction, a 11 min (for LCM) or 15 min heat shock at 37°C was applied 36 hr AEL, after which the L1 larvae were transferred to normal food or NR. For adult fly eye measurement and eclosion assays, crosses were set up on normal food for 3 days followed by transfer of flies to NR for 3 days. Adult flies were analyzed 3 days after eclosion. *CG1315* was used as control for RNAi experiments and *FRT82 iso* was the control for clones generated using mitotic recombination.

## Microscopy and immunofluorescence
Adult females were frozen at −20°C 3 days after eclosion. The flies were mounted on agar plates containing charcoal. The right-side eyes were imaged on a KEYENCE VHX1000 digital microscope.

Eye imaginal discs were dissected 108 hr AEL from normal food and 156 hr AEL from NR. The discs were fixed in 4% paraformaldehyde (PFA, 30 min, room temperature (RT)), washed thrice in 0.3% Triton-X in PBS (PBT, 15 min, RT), blocked in 2% Normal Donkey Serum in 0.3% PBT (2 hr, 4°C), incubated with primary antibodies (overnight, 4°C), washed thrice in 0.3% PBT (15 min each, RT), incubated with secondary antibodies (2 hr, RT), washed thrice in 0.3% PBT (15 min each, RT), stained with DAPI in 0.3% PBT (1:2000, 10 min, RT), and washed once with PBS (10 min, RT). The samples were mounted on glass slides in VECTASHIELD (Vector Laboratories H-1000). Confocal images were obtained on a Leica SPE TCS confocal laser-scanning microscope.

The following antibody dilutions were used: rabbit anti-FoxO (1:500 for immunostaining and 1:9000 for immunoblotting, gift from Mark Tatar [*Bai et al., 2012*]), mouse anti-Cnx99A (1:25, DSHB Cat#Cnx99A 6-2-1), mouse anti-α-tubulin (1:20000, Sigma-Aldrich Cat#T9026) and mouse anti-GFP (1:200, Thermo Fisher Scientific Cat#A-11120). DAPI (1:2000) was used to stain nuclei. Secondary antibodies used were: goat anti-rabbit Alexa Fluor 633 (1:500, Thermo Fisher Scientific), goat anti-mouse Alexa Fluor 568 (1:500, Thermo Fisher Scientific) and goat anti-mouse Alexa Fluor 647 (1:500, Thermo Fisher Scientific).

## Cell culture RNAi and western blot

dsRNA against EGFP (control) or *foxo* was prepared as described earlier (*Okada et al., 2015*) using primers in *Supplementary file 2*. 15 µg dsRNA in 500 µL water was added to each well of a 6-well plate. $3 \times 10^6$ cells in 1 mL serum-free media were added to each well and the plate was incubated at RT for 30 min. 3 mL complete media was added after 30 min. The cells were harvested after 96 hr. 1.5 mL was used for RNA isolation, and proteins were extracted from 3 mL cell suspension. The cells were centrifuged and supernatant was removed. The cells were washed with ice-cold PBS, resuspended in protein extraction buffer (50 mM Tris-HCl (pH 8.0), 120 mM NaCl, 20 mM NaF, 1 mM benzamidine, 1 mM EDTA, 6 mM EGTA, 15 mM $Na_4P_2O_7$, 1% Nonidet P-40, and protease inhibitor cocktail (Roche, 1 tablet in 14 mL buffer)) and incubated on ice for 30 min. Thereafter, the lysates were cleared by centrifugation at 14000 g at 4°C for 15 min.

Protein concentrations were measured by BCA Protein Assay (Thermo Fischer Scientific). Proteins were denatured by boiling for 7 min in sample loading buffer and 40 µg of proteins were analyzed by immunoblotting. Briefly, proteins were resolved by 4–8% SDS-PAGE followed by transfer onto 0.45 µm nitrocellulose membrane. After blocking, primary antibodies were incubated overnight at 4°C and immunoblot signals were detected after secondary antibody incubation using enhanced chemiluminescence (ECL).

## Gene expression analysis

Cell pellets were processed as described in Materials and methods. For eye discs, larvae were dissected in HBSS. 30 discs were collected and frozen in liquid nitrogen. RNA was isolated using RNeasy Plus Micro kit (QIAGEN). After reverse-transcription with SuperScript III First-Strand Synthesis System (Thermo Fisher Scientific), qPCR was performed using LightCycler 480 SYBR Green I Master (Roche) on a Rotor-Gene 6000 (Corbett). Gene expression levels were normalized to the expression level of *RpS23*. Primer sequences are described in *Supplementary file 2*.

## Quantification and statistical analysis

Adult eye area was measured using ImageJ (*Schindelin et al., 2012*). Ratio of nuclear FoxO intensity was also calculated using ImageJ. The mean gray values were measured for five nuclei in the clone and surrounding wild-type tissue from the same disc using the Multi-point tool. The intensity of the clone was divided by the intensity of the wild-type area to get the ratio. Statistical analyses were performed using unpaired two-tailed Student's t-test. *p* values are described in the Figure legends. All plots were generated in R Studio, and Figures were assembled using Adobe Illustrator.

## Acknowledgements

We thank Dr. Hyung Don Ryoo, the Bloomington *Drosophila* Stock Center (BDSC), the Vienna *Drosophila* Resource Center (VDRC), and the Kyoto *Drosophila* Genetic Resource Center for flies, the *Drosophila* Genomics Resource Center (DGRC) for Kc167 cells, and Dr. Mark Tatar and the Developmental Studies Hybridoma Bank (DSHB) for antibodies. We are grateful to Joachim Hehl at

ScopeM ETH Zurich for help with LCM, Functional Genomics Center Zurich for RNA sequencing, Dr. Michal Okoniewski at IT-SIS ETH Zurich for help with sequencing data analysis, Dr. Aurelio Teleman for discussing FoxO targets in flies, students from the 2018 Growth Control Course for assisting in experiments, and Angela Baer and Igor Vuillez for technical assistance.

## Additional information

### Funding

| Funder | Grant reference number | Author |
|---|---|---|
| Schweizerischer Nationalfonds zur Förderung der Wissenschaftlichen Forschung | SNF 31003A_166680 | Hugo Stocker |
| Krebsliga Schweiz | KLS-3407-02-2014 | Hugo Stocker |
| Novartis Stiftung für Medizinisch-Biologische Forschung | 18C162 | Hugo Stocker |

The funders had no role in study design, data collection and interpretation, or the decision to submit the work for publication.

### Author contributions

Avantika Gupta, Conceptualization, Formal analysis, Investigation, Visualization, Methodology, Writing - original draft; Hugo Stocker, Conceptualization, Supervision, Funding acquisition, Methodology, Writing - original draft

### Author ORCIDs

Avantika Gupta [iD] https://orcid.org/0000-0002-8170-6720
Hugo Stocker [iD] https://orcid.org/0000-0002-4142-1464

### Decision letter and Author response

Decision letter https://doi.org/10.7554/eLife.53159.sa1
Author response https://doi.org/10.7554/eLife.53159.sa2

## Additional files

### Supplementary files

- Supplementary file 1. Key Resources Table.
- Supplementary file 2. Primer sequences for dsRNA synthesis and qPCR.
- Transparent reporting form

### Data availability

Sequencing data have been deposited in GEO under accession code GSE138877.

The following dataset was generated:

| Author(s) | Year | Dataset title | Dataset URL | Database and Identifier |
|---|---|---|---|---|
| Gupta A, Stocker H | 2019 | FoxO targets in Tsc1-deficient cells under normal food and nutrient restriction food | https://www.ncbi.nlm.nih.gov/geo/query/acc.cgi?acc=GSE138877 | NCBI Gene Expression Omnibus, GEO GSE138877 |

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
