## [Decision Letter]

**Acceptance summary:**

This study examines the genetic interaction between *Tsc1* and FoxO in a *Drosophila* tumor model. This builds on previous work showing that when *Drosophila* larvae are raised under nutrient restrictive conditions, simultaneous loss of both *Tsc1* and FoxO results in an overgrowth phenotype. Data are provided here that in a *Tsc1* mutant background, FoxO limits cell proliferation to restrict mutant overgrowth. The study also finds down-regulation of genes associated with the ER stress response pathway in FoxO deficient cells and knockdown of putative FoxO target genes in Tsc1-deficient cells also induces ectopic growth. This leads to the conclusion that the ER stress response pathway is a key FoxO-regulated pathway that impacts growth control downstream of Tsc1.

**Decision letter after peer review:**

Thank you for submitting your work entitled "FoxO suppresses endoplasmic reticulum stress to inhibit growth of Tsc1-deficient tissues under nutrient restriction" for consideration by *eLife*. Your article has been reviewed by three peer reviewers, and the evaluation has been overseen by a Reviewing Editor and Utpal Banerjee as Senior Editor. The reviewers have opted to remain anonymous.

The manuscript has some issues that need to be addressed before acceptance. As you will see from the reviewer comments, all found the work to be interesting, but also raised some points that we felt would be straight forward to address in a revised manuscript.

Reviewer #1:

This study by Gupta and Stocker uses a transcriptomics approach to examine the genetic interaction between *Tsc1* and FoxO in a *Drosophila* tumor model. Previous studies in the fly demonstrated that when *Drosophila* larvae are raised under nutrient restrictive conditions, simultaneous loss of both *Tsc1* and FoxO in the eye imaginal disc results in an overgrowth phenotype. In an effort to identify the FoxO target genes that contribute to this phenotype, they use laser capture microdissection combined with RNAseq to identify those genes that are differentially regulated in cells lacking FoxO. The resulting analysis reveals a significant down-regulation of genes associated with the ER stress response pathway in FoxO deficient cells. Moreover, these genes appear to be important for the FoxO/*Tsc1* phenotype because RNAi of these putative FoxO target genes in Tsc1-deficient cells also induced ectopic growth. Similarly, the use of small molecules to induce ER stress enhanced the size of eyes lacking Tsc1.

Overall, I found this to be a short, but potentially interesting manuscript. The use of mosaic analysis combined with laser capture microdissection for conducting RNA-seq is a powerful approach. In addition, I was impressed by the fact that down-regulation of the ER stress response enhance the *Tsc1* overgrowth phenotype, suggesting that this is a key FoxO-regulated pathway in the context of growth control. However, I have a few criticisms that should be addressed:

1) The text repeatedly refers genes mis-regulated in the RNAseq analysis as FoxO targets. However, there is no FoxO binding analysis to prove such an assertion and the text should be modified accordingly.

2) Several of the genes that are classified as being part of the ER stress response have not been characterized in flies (e.g., CG6766). The authors should verify that RNAi-depletion of these candidate genes induce ER stress using either the Cnx99a antibody or by examining expression of spliced Xbp1 transcripts.

3) In paragraph two of the Results you speculate about the reason why snoRNAs are up-regulated in the RNAseq data. Since there are no experiments examining why snoRNAs are mis-regulated in the dataset, most of this text should be moved to the Discussion.

Reviewer #2:

This paper follows on from a previous report from the Stocker lab showing that *foxo* normally functions to limit *tsc1* mutant epithelial cell clone growth and when *foxo* is removed, these clones show massive overgrowth. The interpretation from this work was that, in a *tsc1* mutant background, *foxo* may be functioning to limit cell proliferation to restrict *tsc1* mutant overgrowth. In this paper, Gupta and Stocker now use transcriptomics to identify FoxO regulated genes that may explain this function of FoxO.

They use a very nice LCM technique to isolate cell clones, which presumably removes potential caveats and confounds in expression data associated with tissue disruption/FACS sorting, which is often used as a way to isolate pure populations of cell clones from imaginal discs. In addition, the finding that alteration in ERAD genes may explain the effects of FoxO on growth and proliferation of *tsc1* mutant cells is a novel finding that will be of broad interest to researchers in the field of cell growth and proliferation (not just in flies).

1) Results, paragraph one: It's strange that the RNA-seq showed no FoxO knockdown. The conclusion is that the lack of detection of FoxO knockdown is due to biological variability. And the authors show an example of partial mRNA knockdown achieving strong protein knockdown in Kc167 cells. However, the eye disc data show no knockdown at all. Is it possible that the LCM technique may not be accurately just isolating the knockdown clones? Based on the schematic in Figure 1, can the authors use presence/absence of GFP transcripts to clearly delineate that have isolated purely the clonal cells?

2) The discovery that regulation of ER stress and ERAD genes may explain the effects of FoxO is interesting. These finding would certainly be in line with the known links between Tsc1, FoxO and the control of ER stress. One thing I thought was missing was a little more investigation and explanation of how the ER stress genes are interacting with Tsc1 to control cell proliferation – the data in Figures 2, 3 do show changes in eye size, but is it clear that these effects are on cell proliferation (vs. growth, death)? And if they are, what might be the mechanism? As it stands the paper reads as an initial (but still very nice) functional validation of RNA-seq data rather than an in-depth mechanistic investigation into how FoxO and Tsc1 function to control tissue growth.

Reviewer #3:

This manuscript is a huge technical achievement, applying cutting edge clonal techniques to identify FoxO target genes in *Drosophila*. The results reveal regulation of the ER stress response by FoxO along with other targets. The model system used is excellent, because it enables full FoxO activation in growing clones of cells in the eye, which is achieved by mutating TSC to activate a TOR negative feedback loop to suppress Akt, thereby activating FoxO. This roundabout method is essential because simply blocking Akt or over activating FoxO normally kills cells. Overall, I found the results robust, important and of interest to the field.

1) Title could be shorter. No need to mention TSC.

2) Introduction ought to more carefully and fully explain the TOR negative feedback loop on Akt that means full FoxO activation upon loss of TSC.

---

## [Author Response]

Reviewer #1:[…] However, I have a few criticisms that should be addressed:1) The text repeatedly refers genes mis-regulated in the RNA-seq analysis as FoxO targets. However, there is no FoxO binding analysis to prove such an assertion and the text should be modified accordingly.

The reviewer is right. We have modified the text to correctly refer to those genes as genes downregulated in *Tsc1* mutant cells upon *foxo* knockdown.

2) Several of the genes that are classified as being part of the ER stress response have not been characterized in flies (e.g., CG6766). The authors should verify that RNAi-depletion of these candidate genes induce ER stress using either the Cnx99a antibody or by examining expression of spliced Xbp1 transcripts.

We investigated the induction of ER stress response in cells mutant for the top candidate CG6766. We used a mutant allele carrying a piggyBac insertion with a splice-acceptor site and stop codons in all frames in the first intron of the gene (Schuldiner et al., 2008; line 140290 from Kyoto *Drosophila* Genetic Resource Center). Calnexin staining of homozygous mutant clones showed similar staining pattern compared to the surrounding wild-type tissue and the isogenic control clones (Figure 2—figure supplement 2A). We then examined the expression levels of spliced-*Xbp1* and *Hsc-70-3* to observe the effect on the upregulation of ER stress response. Control or *CG6766* knockdowns were generated in wild-type, weak *Tsc1* knockdown and strong *Tsc1* knockdown backgrounds. qRT-PCR analysis on eye imaginal discs revealed no change in the expression levels of spliced-*Xbp1* or *Hsc-70-3* in the wild-type background upon knockdown of *CG6766* (Figure 2—figure supplement 2B and B'). By contrast, the expression levels of both ER stress reporters were increased in the *Tsc1* knockdown backgrounds, indicating that CG6766 is limiting to maintain ER homeostasis in *Tsc1* mutant cells.

The lack of an ER stress response in cells mutant for CG6766 suggests a (partially) redundant function. This may also explain why CG6766 has not been characterized so far.

3) In paragraph two of the Results you speculate about the reason why snoRNAs are up-regulated in the RNAseq data. Since there are no experiments examining why snoRNAs are mis-regulated in the dataset, most of this text should be moved to the Discussion.

We agree with the reviewer. The text relating to the upregulation of snoRNAs has been moved to the Discussion.

Reviewer #2:[…]1) Results paragraph one: It's strange that the RNA-seq showed no FoxO knockdown. The conclusion is that the lack of detection of FoxO knockdown is due to biological variability. And the authors show an example of partial mRNA knockdown achieving strong protein knockdown in Kc167 cells. However, the eye disc data show no knockdown at all. Is it possible that the LCM technique may not be accurately just isolating the knockdown clones? Based on the schematic in Figure 1, can the authors use presence/absence of GFP transcripts to clearly delineate that have isolated purely the clonal cells?

We were also puzzled by the unaltered levels of *foxo* mRNA in the *foxo* RNAi samples. In addition to the biological variability, we suspect the presence of a transcriptionally-activating feedback upon *foxo* knockdown as mentioned in the manuscript. An effect on translational capacity could also explain the loss of FoxO protein (Figure 1B and 1B’) with no effect on the transcript levels (Figure 1 —figure supplement 1B).

The reviewer is concerned that contamination with non-clonal tissue could account for the high *foxo* mRNA levels. We don’t think that contamination is a major issue in our experiments. The most straightforward test for a contamination of the clonal tissues would be to measure the GFP expression levels (as suggested by the reviewer). Unfortunately, a direct comparison is not possible because, in order to maximize the number of biological replicates, we did not sequence GFP-positive clones. However, we compared the *GFP* expression levels in cells with and without *foxo* knockdown. As an example, the expression level in *Tsc1* mutant clones isolated from normal feeding conditions is shown in Author response image 1. As no major differences were observed, the contamination from GFP-positive cells, if any, is similar in all conditions. Additionally, representative images of isolated *Tsc1* mutant clones with and without *foxo* knockdown are presented to support that the isolated cell populations were negative for GFP (Figure 1—figure supplement 2A). This is further substantiated by the presence of 100% mutant transcripts (*Tsc1^Q87X^* where **C**AG at position 3R:24,130,984 is substituted by **T**AG) in the *Tsc1* mutant clones (Figure 1—figure supplement 2B). Taken together, it is very unlikely that the isolated clonal samples were contaminated with surrounding non-clonal cells.

**Author response image 1. respfig1:** *GFP* expression levels in *Tsc1* mutant clones with and without *foxo* knockdown isolated from larvae raised under normal feeding conditions. Data are represented as mean ± SD.

2) The discovery that regulation of ER stress and ERAD genes may explain the effects of FoxO is interesting. These finding would certainly be in line with the known links between Tsc1, FoxO and the control of ER stress. One thing I thought was missing was a little more investigation and explanation of how the ER stress genes are interacting with TSC1 to control cell proliferation – the data in Figures 2, 3 do show changes in eye size, but is it clear that these effects are on cell proliferation (vs growth, death)? And if they are, what might be the mechanism? As it stands the paper reads as an initial (but still very nice) functional validation of RNA-seq data rather than an in-depth mechanistic investigation into how FoxO and Tsc1 function to control tissue growth.

To address if the *Tsc1* mutant tissue overgrowth upon induction of ER stress (as depicted in Figures 2 and 3) is due to increased proliferation or cell size, we analyzed the density of ommatidia in adult eyes. Since the loss of Tsc1 does not disrupt ommatidial architecture (Nowak et al., 2018), the density of ommatidia reflects the cellular density in the adult eyes. This density remains unchanged between *Tsc1* and *Tsc1 foxo* knockdown eyes but is slightly reduced upon knockdown of *CG6766* and *CG8860* (Author response image 2), indicating that at least a part of the enhanced overgrowth of the adult eyes is due to an increase in cell size. However, the small reduction in density (10% for *CG6766* and 7% for *CG8860*, respectively) cannot fully account for the observed overgrowth (Figure 2B, 23% average increase for *CG6766* knockdown and 11% average increase for *CG8860* knockdown). We therefore conclude that the overgrowth is in part caused by more cells.

**Author response image 2. respfig2:** 

In the Discussion, we speculate about the mechanisms of how ER stress response pathways can regulate proliferation. These are reinforced by the results presented in Figure 4.

We agree with the reviewer that our manuscript represents an initial analysis to describe a link between FoxO and ER stress response. We are confident that it will serve as a strong foundation for future studies aiming at elucidating the exact mechanism of regulation of ER stress by FoxO and the effects on growth.

Reviewer #3:[…]1) Title could be shorter. No need to mention TSC.

We are aware that the title is rather long and complicated. However, the identification and analysis of differentially regulated genes were based on clonal populations of *Tsc1* mutant cells. The manuscript does not address a generalized function of FoxO in regulating ER stress.

In our opinion, it is more accurate to mention in the title that our findings are specific to Tsc1-deficient tissues.

2) Introduction ought to more carefully and fully explain the TOR negative feedback loop on Akt that means full FOXO activation upon loss of TSC.

The text in the Introduction has been revised to explicitly state the negative feedback to insulin signaling in hyperactive mTORC1 cells and to describe the nuclear localization and activation of FoxO in *Tsc1* mutant cells.